# Developing Green Healthcare Activities in the Total Quality Management Framework

**DOI:** 10.3390/ijerph19116504

**Published:** 2022-05-26

**Authors:** Sang M. Lee, DonHee Lee

**Affiliations:** 1College of Business Administration, University of Nebraska-Lincoln, Lincoln, NE 68588, USA; slee1@unl.edu; 2College of Business Administration, Inha University, Incheon 22212, Korea

**Keywords:** green healthcare, continuous improvement activities, total quality management framework, healthcare industry

## Abstract

This study examines the effectiveness of green healthcare activities in hospitals based on the total quality management (TQM) framework. The proposed research model and associated hypotheses were tested using structural equations modeling based on the data collected from 261 employees at general hospitals in South Korea. The results of the study revealed that the role of top management is essential for the successful implementation of green healthcare activities through motivating employees for their active participation in the program, providing continuous education and training on the importance of environmental sustainability, and diligent monitoring of the progress at the organization level. The study findings provide theoretical and practical implications on strategic approaches to planning and implementing green healthcare activities in hospitals for the greater good.

## 1. Introduction

The current business environment is volatile due to the global pandemic, geopolitical conflicts, climate change, chemical pollution, and the overuse of limited resources [1]. Climate change is especially troublesome, as it adversely affects human health, leading to an increase in associated diseases that impose pressure on healthcare systems [2]. The World Health Organization (WHO) has emphasized the need to reinforce the public health system, emergency response programs, and relevant research for a sustainable environment. Specifically, the healthcare sector is expected to play a crucial role in mitigating the effects of climate change on human health [3].

The impacts of the healthcare industry on humanity and the environment stem from the resource-intensive nature of the industry [4]. Healthcare organizations are representative energy-consuming institutions (e.g., 9–10% of greenhouse gas emissions originate from the healthcare sector in the United States) because they consume large quantities of disposable products and generate an enormous amount of toxic waste that contributes to environmental pollution [5,6]. Under this context, the concept of ‘green healthcare’ was introduced by WHO [7].

In 2000, the U.S. Green Building Council (USGBC) announced a green building certification program called Leadership in Energy and Environmental Design (LEED) [8]. Subsequently, healthcare organizations have been engaged in activities to mitigate risk factors of unsustainable development/production to pursue a sustainable healthcare/treatment environment [4]. In 2004, the European Union (EU) adopted the Vienna Declaration and has since implemented eco-friendly policies, such as reducing the use of polyvinyl chloride (PVC) in healthcare facilities, using alternative energy sources, and purchasing eco-friendly equipment/material. In Brazil, hospitals consume 10.6% of the energy used for commercial purposes [9]. The National Health Service (NHS) of the United Kingdom emits 18 million tons of CO_2_ per year, with nearly a quarter of the total emissions originating from the public sector [10]. In the US, the overall gas emission by healthcare organizations increased 6 percent from 2010 to 2018 [11]. India generated over 33,000 tons of medical waste during the seven months of the COVID-19 pandemic [12]. Furthermore, the scale of the global medical waste management market is estimated to grow from USD 6.8 billion in 2020 to 9 billion by 2025 [13].

Healthcare organizations have been striving to maintain high-quality care services, prevent the spread of diseases, and sustainably manage hospitals [14,15]. Simultaneously, to address the energy use, appropriate waste disposal, and maintain a hospital environment that minimizes risks to patients and local communities, they have adopted various environmental management programs related to green healthcare [4,14]. The need for green healthcare has steadily gained greater recognition as hospitals are resource-intensive organizations that are using an increasing amount of public resources (e.g., water, gas, electricity, etc.), food, and facilities to provide medical services [4]. Therefore, in the current context, where climate change and viruses threaten humanity and the natural environment, it is imperative that the healthcare sector need to expand its investment in environmental protection initiatives to reduce waste generation by implementing sustainable practices.

Green healthcare aims to concomitantly minimize negative environmental impacts and eradicate diseases by recognizing the relationship between human and environmental health [16]. Green healthcare also encompasses the concept of eco-friendliness, denoting that it provides eco-friendly care services that aim at not only promoting personal health, but also positively affecting the community [17]. Furthermore, green healthcare can create economic value by reducing waste and operational costs, increasing the value of healthcare facilities, and improving consumer awareness about the importance of sustainability. Additionally, it has the goal of achieving the greater good through supporting the creation of a sustainable ecosystem [17]. However, despite these advantages, the implementation of green healthcare entails high-level hardware requirements, such as green infrastructure components for the hospital. Given this scenario, various studies have been conducted on the related topics, such as research on the evaluation criteria for hospital construction certification [4,14,18], case studies on minimizing the environmental impact of patient treatment [19], and theoretical analyses on the topic [20]. However, there is a paucity of empirical research on green healthcare practices.

Total quality management (TQM), as an innovative management method for continuous improvement, emphasizes customer satisfaction, education and training, job-related processes (e.g., guidelines), the role of related departments (e.g., marketing and operation management), and participation of all employees [21,22,23]. By considering TQM as the framework for devising operational plans of green healthcare, the participation and commitment of all employees are crucial. Accordingly, this study intends to expand the scope of research on the implementation of green healthcare by incorporating TQM principles into the corresponding operational strategies.

This study first conducts a literature review to derive a theoretical framework based on the TQM and green healthcare perspectives. The purpose of this study is to examine the role of the top management in the participation of employees in green healthcare activities, education and training, and monitoring operations/systems. Additionally, this study empirically tests the relationships among these factors and continuous improvement activities associated with green healthcare, apart from the relationship between continuous improvement activities and environmental performance. Data were collected from hospitals in South Korea, and the proposed model will be tested using a structural equation modeling (SEM) approach. The rest of the paper is organized as follows: Section 2 presents the literature review; Section 3 provides the research model and hypotheses; Section 4 describes the methodology; Section 5 reports the results; and Section 6 presents the conclusions and limitations of the study.

## 2. Literature Review

### 2.1. Green Healthcare

The importance of sustainability was declared at the 1972 United Nations Conference on Human Environment in Stockholm, Sweden [24]. Sohn [25] presented sustainability as “the protection and improvement of the human environment is a major issue which affects the well-being of people and economic development throughout the world, it is the urgent desire of the people of the whole world and the duty of all governments.” Correspondingly, the Brundtland report noted that the ‘environment’ and ‘development’ are non-separable entities as they are interrelated in a causal system [26]. This report presented sustainable development as “a process of change in which the exploitation of resources, the direction of investments, the orientation of technological development, and institutional change are all in harmony and enhance both current and future potential to meet human needs and aspirations”. Elkington [27] further described a triple bottom line (TBL) approach to achieving sustainability based on economic, environmental (or ecological), and social responsibility factors.

Since its declaration in 1972, sustainability has been widely applied to the healthcare industry (i.e., green healthcare and green hospitals) [4], with researchers and organizations defining green healthcare in various ways. Howard [16] introduced the following definition in a report of the Office of the Federal Environmental Executive: “the practice of increasing the efficiency with which buildings and their sites use energy, water, and materials, and reducing building impacts on human health and the environment, through better siting, design, construction, operation, maintenance, and removal”. Kreisberg [28] explained that green healthcare facilitates a sustainable future for medicine, physicians, patients, and the environment. These views emphasize that green healthcare plays a critical role in improving the health of people, communities, and the environment. Taleshi et al. [29] suggested that the practice of green healthcare is an enabler of a healthy life by reducing the environmental impact and taking responsibility for sustainable disease treatment activities.

The Green Guide for Health Care [30] announced eco-friendly elements for establishing green hospitals (see Table 1). Meanwhile, as the global impacts of climate change are expected to be severe for some and catastrophic for others, WHO [3] issued a declaration for eco-friendly policies. WHO [3] emphasized the importance of developing a climate-friendly, cost-saving strategy and proposed seven implementation dimensions that could enable common health, environmental sustainability, and social benefits. In 2017, IOM presented action plans to build a sustainable environment through the Environmental Sustainability Programme (ESP). In this program, several categorized areas of environmental management are suggested, such as water, energy, and waste disposal [31].

Amid several green healthcare initiatives, the Environmental Excellence Award (EEA), provided by Practice Greenhealth [32], is the highest honor that healthcare organizations can receive for their green-related activities. The award is bestowed upon those who lead a global movement for environmental health and justice by minimizing the environmental footprint through innovative healthcare services and establishing a sustainable operational environment [32]. The EEA has been awarded yearly to 25 healthcare organizations since 2016 and is divided into 10 categories: “leadership, waste, chemicals, greening the OR (operating rooms), food, environmentally preferable purchasing, energy, water, climate, and green building”. The top 10 hospitals in each category are also selected [32].

Additionally, green healthcare-related organizations and associations have developed metrics for assessing green healthcare activities (Table 1). Although there is ongoing research on these aspects, they have yet to provide verified and systematized findings. For example, Dhillon and Kaur [4] analyzed research works on green healthcare through Google search engine. This analysis yielded seven indicators for assessing the green activities of hospitals: energy conservation, alternative means of energy generation, designing green buildings, waste management, water conservation, reducing transportation costs, and providing healthy food. Azar et al. [33] conducted an empirical analysis of educational programs at teaching and private hospitals, which yielded eight dimensions. Furthermore, Shaabani et al. [14] studied hospitals in Iran and suggested nineteen dimensions of green healthcare activities be implemented in accordance with governance and social responsibility. As medical institutions begin their sustainability journey with different organizational conditions, some may quickly develop feasible and effective practices, while others need to devote continuous efforts towards the same goal [34].

Various hospitals have been striving towards green healthcare, including the provision of organic food and seasonal menus, activities to reduce hospital waste (as observed in Bethesda Hospital, Hamburg, Germany), and efforts to eliminate the use of PVC products (as observed in Karolinska University Hospital, Solna, Sweden) [35]. In 2017, an EEA winner, the Mayo Clinic in Eau Claire, Wisconsin, saved enough water to fill 50 Olympic-sized swimming pools and 25% of its energy use was derived from renewable resources. The hospital implemented green activities through a program that included reusing 3.3 tons of surgical tools and 7.3 tons of plastics, recycling 2.9 tons of batteries, and composting food waste [36]. Moreover, the Mayo Clinic actively encourages its employees to participate in environmental activities, such as campaigns to demonstrate how recycling aluminum can conserve energy equivalent to three hours of computer use or two hours of television watching, and how recycling one glass bottle saves the same energy required for using a 100-watt lightbulb for four hours. Tennison et al. [37] cited the sources of healthcare PVC pollutants from the UK National Health Service Report that “62% came from the health-related supply chain (e.g., medical equipment, non-medical equipment, pharmaceuticals and chemicals, food and catering, business services, and other procurement), 24% from the direct delivery of care, 10% from staff commute and patient and visitor travel, and 4% from private health and care services commissioned”. Overall, green healthcare can be implemented in facilities including medical offices, clinics, and small-sized hospitals to large healthcare organizations. Considering the diverse application areas of green healthcare, there is a need to adopt eco-friendly practices across the healthcare delivery system [17].

The EEA evaluation parameters for green healthcare-related activities are summarized in Table 1. Items 1–5 cannot be adopted without the willingness of the top management. Likewise, items 6–13 cannot be operationalized without the participation of employees and their actions (activities). Therefore, these 13 items can be categorized into two dimensions: the role of top management and the practices of employees.

**Table 1 ijerph-19-06504-t001:** Dimensions of green healthcare.

Related Institutions	Dimensions of Green Healthcare												
	Role of Top Management					Practices of All Employees							
	①	②	③	④	⑤	⑥	⑦	⑧	⑨	⑩	⑪	⑫	⑬
BEPHS [38]				V	V	V	V	V				V	V
EEA [39]	V	V	V		V	V	V	V	V	V		V	
GGHC [40]	V	V		V		V	V		V	V	V	V	V
GGHH [41]		V		V		V	V	V	V		V	V	V
IOM [42]						V	V	V					
ISO 14000 [43]								V	V		V		
PAHO [44]		V		V	V	V	V	V	V				
SHT [45]				V	V	V	V	V	V	V			
USGBC LEED [46]		V	V	V	V	V	V		V	V			
WHO [47]		V		V		V	V	V			V	V	

① Leadership ② Designing green buildings ③ Design and innovation of hospital spaces ④ Environmental management ⑤ Indoor environmental quality ⑥ Energy efficiency ⑦ Water efficiency ⑧ Waste management ⑨ Procurement ⑩ Materials/Resources ⑪ Transportation ⑫ Food ⑬ Management (e.g., patient care, infection control, and laundry).

### 2.2. TQM in Healthcare

Healthcare has become increasingly important during the COVID-19 pandemic as it is a key factor for economic recovery, welfare of people, and return to normal activities of organizations [48]. In addition, during the pandemic, patients are becoming both customers of a healthcare organization and direct strategic partners in the decision-making process. Furthermore, complex current issues involved in environmental, social, governance, and political policies have significant impacts on hospital management, especially the quality of services provided. Therefore, the provision of quality care services is an essential requirement of healthcare organizations [22,48]. The development of such concepts as total quality management (TQM) and Six Sigma has a long history in the field of business administration, and these approaches have been widely implemented in various organizations, including healthcare providers.

Previous research on TQM in healthcare has focused on care services using various quality models (e.g., MBHCP, EFQM, and ISO 9000 series, SERVQUAL, SERVPERF, HEALTHQUAL, etc.). In healthcare services, TQM has been defined in various ways. The American Society for Quality [49] describes TQM as “a management system for a customer-focused organization that involves all employees in continual improvement”. Donabedian [50] defined TQM in healthcare as the “maximization of patient’s satisfaction considering all profits and losses to be faced in a healthcare procedure”. Øvretveit [51] suggested that “TQM is a comprehensive strategy of organizational and attitude change for enabling personnel to learn and use quality methods, in order to reduce costs and meet the requirements of patients and other customers”. Lee and Lee [48] stated that TQM can be achieved by participation in improving processes by all members of an organization. Based on previous studies, Lee and Lee [48] suggested five key components of TQM: The role of leadership, the role of the quality department, employee participation, education and training, and process and operational procedure. Thus, hospitals should develop strategies by integrating quality discipline into their own culture to identify and prioritize activities that help meet patients’ needs and demands.

TQM as a management approach means continuous healthcare quality improvement. For healthcare quality improvement, the organization should focus on preventing medical errors and administrative problems, improving patient and employee satisfaction, and continuously improving work processes and environments [22]. Each hospital has unique own characteristics in terms of its organizational culture, management practices, and the processes to create and deliver care services. Thus, TQM can be a driver that assures quality care services as the outcome of committed employees (e.g., medical staff and administrators) in healthcare organizations [22,48,51,52].

### 2.3. TQM and Green Healthcare

Within the healthcare industry, the TQM paradigm has been widely applied as a management philosophy or strategy [21,22]. The TQM paradigm has been used to facilitate the following: describe the overall activity of a healthcare organization, improve the quality of medical services and customer satisfaction based on the support of the top management, and provide operational strategies for the continuous improvement of all employees [22,53]. The primary components of TQM activities include the role of top management, the participation of employees, education and training, process management and operation procedures, and continuous improvement activities [22,53,54].

Implementation of sustainable corporate activities is impossible without the willingness, interest, and support of top management. According to Kiesnere and Baumgartner [55], around 90% of organizations with the best performance indicate that the role of top management is “a key success factor for the sustainable development of the company”. In particular, environmental, social, and governance (ESG) activities require organization-wide improvement and participation rather than isolated operational instances [55]. For example, in healthcare services, ESG standards can be applied to many areas of the organization, such as energy and waste management, investment in community health, diversity, equity, and inclusion initiatives, which require the willingness and participation of all employees, supported by the top management.

TQM activities can be implemented according to the needs of the organization to drive blueprints (e.g., the vision and strategic plans of the organization) [22,23]. In addition, TQM can help improve the quality of care and customer satisfaction while reducing waste with the participation of all employees. Accordingly, most organizations where top management is committed to pursuing quality improvement through TQM activities tend to achieve positive results [22,23]. Another important benefit of TQM activities is that they provide an opportunity to improve the organizational structure and operational processes of the organization [21]. Hence, if a healthcare organization is committed to applying TQM principles, major initiatives related to TQM can also be implemented for green healthcare. Examples of green healthcare activities include the construction of green buildings to reduce energy consumption, use of renewable energy, and creation of a green environment to help improve patient recovery. In summary, the perspectives of TQM and management innovation may be applicable to, and concordant with, green healthcare activities.

## 3. Research Model and Hypotheses Development

### 3.1. Research Model

In accordance with the above reviewed research, this study developed operational strategies for green healthcare implementation from a TQM perspective. It was assumed that the role of top management has an impact on the participation of employees in green activities, education and training, monitoring of the activities of employees, and continuous green healthcare improvement, which can enhance environmental performance. The proposed research model is presented in Figure 1.

The role of top management is to increase their employees’ motivation and morale by setting and delivering viable visions and goals that support the organization’s long-term prosperity [22]. WHO [3] emphasized that the healthcare industry should respond to climate change by playing a moral/ethical and practical leadership role through its green practices. Accordingly, top management in healthcare organizations should show its leadership for green practices, since sustainability generally requires a change in organizational culture, which needs to be supported by appropriate policies, resources, and visions [19,34]. Furthermore, top management is responsible for ensuring that the implementation of green healthcare is aligned with organizational philosophies and goals, thereby securing long-term application of and support for such activities [14,34].

Green healthcare providers should also improve both the infrastructural aspect of their operations and the design aspect of their divisional layout [33]. Altomonte et al. [18] argued that the effectiveness of green strategy to enhance patient satisfaction requires continuous monitoring of green activities and evaluation of user feedback. For the implementation of green healthcare to be successful, strong leadership of top management is imperative [33]. Thus, top management should be the driving force for employing the green healthcare strategy by sharing the vision with all employees through authentic leadership, motivating employee participation, providing education and training opportunities, and monitoring activities. Therefore, the following hypotheses are proposed.

**Hypothesis** **1** **(H1).**
*The role of top management has a positive effect on the participation of employees in green healthcare activities.*


**Hypothesis** **2** **(H2).**
*The role of top management has a positive effect on the education and training of employees in green healthcare.*


**Hypothesis** **3** **(H3).**
*The role of top management has a positive effect on the monitoring activities/systems of employees related to green healthcare.*


Positive employee participation should be the basis for achieving organizational goals [22]. However, as emphasized previously, the efforts of only one organizational unit (e.g., top management, the care department, or administrative support departments) would not be sufficient to ensure the successful implementation of green healthcare [56]. Instead, it is essential to secure the participation and organic collaboration of various healthcare staff and departments and cooperation among patients, guardians, and business partners (e.g., medicine and medical support suppliers) [57]. Thus, an approach that integrates all the factors affecting the healthcare environment is required for an organization to ensure the successful implementation of green healthcare activities [20,56]. Shen et al. [58] also stated that green goals of an organization can be achieved when employees fully buy into the program. It is, therefore, important to encourage employees to engage in green initiatives consistent with the organizational vision [59,60]. If the organization is responsible for implementing its vision for environmental management through green initiatives, it is necessary to provide its employees authority and responsibility to fully participate in green activities [61]. In addition, employees should be empowered and encouraged to operate their work in accordance with the organization’s green objectives [58,61]. As shown in Table 1, the metrics of green healthcare are related to food, energy, water, waste, medical supplies, purchasing, and transportation, each of which requires the interests and efforts of all employees [20,56,57,62]. As such, the active participation of employees is essential for implementing successful green healthcare activities.

To achieve organizational goals related to the successful implementation of TQM, continuous education and training and the development of proper human resources for essential knowledge, skills, consciousness, and beliefs are required [22,63]. In terms of green healthcare, to ensure a participatory learning atmosphere and successful job performance, all necessary support should be made available to employees. Specifically, various pertinent education and training opportunities should be provided to employees, including green healthcare-related education opportunities from major international agencies such as green healthcare information, global policies, information on disruptions of natural resources (e.g., water and materials), community pollution, and ethics [20]. Therefore, education and training for green healthcare are likely to have a positive effect on green healthcare improvement activities.

The management of processes and operations refers to the standardization and methods required for individuals to achieve successful work performance, including procedures and systems, organizational structures, and operational processes [22]. Healthcare organizations should develop green awareness through training and education to help employees understand green concepts. This preparation is imperative for developing the basic skills necessary to implement green concepts and effectively achieve green management goals [61]. Green healthcare-based practices are crucial for optimizing the benefits for the organization [19,20]. For example, as healthcare organizations discharge substantial amounts of medical wastes, they can practice green healthcare by continuously monitoring hazardous chemicals that they use and dispose [64,65], as well as adopting recycling strategies to reduce the waste volume and disposal costs [66]. These activities can lead to more consistent practices and the standardization of work performance and operational processes [20]. Moreover, they can have a positive impact on the overall organizational performance by effectively reducing costs and waste through the continuous evaluation of green healthcare activities [64]. Thus, monitoring green healthcare activities in each department could have a positive impact on performance and lead to improvements in an organization.

Therefore, participation of employees, education and training, monitoring of the activities of employees in green activities would positively impact continuous green healthcare improvement. Thus, the following hypotheses are suggested:

**Hypothesis** **4** **(H4).**
*The participation of employees in green healthcare activities has a positive effect on continuous improvement related to green healthcare.*


**Hypothesis** **5** **(H5).**
*Education and training of employees in green healthcare activities have a positive effect on the continuous improvement related to green healthcare.*


**Hypothesis** **6** **(H6).**
*The monitoring activities/systems of employees in green healthcare activities have a positive effect on the continuous improvement related to green healthcare.*


In the healthcare field, continuous improvement activities refer to processes that account for the need to supplement/improve initial goals and activities to ensure that they align with changes in organizational values. The latter, in turn, generally change owing to modifications in the internal (e.g., work improvement) and external (e.g., customers and market dynamics) environments related to the fluid nature of healthcare delivery processes and customer demands [22]. As aforementioned, the continuous improvement activities in TQM may be synonymous with those in green healthcare, as they both can be achieved through the combined, long-term efforts of all employees. Marimuthu and Paulose [20] presented four key elements of operating processes for the emergence of green healthcare in organizations: “environment concerns, the needs of patients, needs of employees, and community concern to continuously improve the quality and reduce cost”. Moreover, they stressed that to ensure the highest quality of services at minimal cost, these factors should be continuously evaluated and improved. Healthcare organizations should ensure that employees contribute to the green goals by appropriately evaluating the green behavior of its employees, aligning this behavior with appropriate incentives for opportunities, pay, and compensation, and encouraging and motivating them to be fully committed to green activities [61,67]. Thus, green healthcare activities cannot be short-term and on–off activities, as they require long-term continuity and assessment. Consequently, medical institutions may achieve better results by implementing continuous improvement activities to enhance their green healthcare operations.

Healthcare organizations are in a high energy-consuming industry [5,6]. Moreover, any policy for reducing global carbon emissions would possibly include a clause on the imposition of carbon taxes, directly affecting the operational cost of the organization. In a study on the role of the healthcare industry in the global climate crisis, Eckelman and Sherman [5] found that traditional assessments of healthcare systems did not consider the costs of overall environmental pollution, ranging from resource extraction to waste management. Kalantary et al. [68] reported that medical waste increased to about 102.2% during the COVID-19 pandemic period compared with the pre-COVID-19 period in Iran. What is the impact of this growth rate on the environment? It may be impossible to financially express the degree of impact on the economy and community; thus, the negative impact of medical waste on the environment will be difficult to estimate.

As shown in Table 1, organizations related to green healthcare suggest the need to incorporate various dimensions of eco-friendly policies during implementation, such as recycling, waste reduction and management, water conservation, PVC reduction, use of eco-friendly foods and materials, construction of green buildings, use of alternative energy, and purchase of eco-friendly products. These green healthcare practices can be applied across most operations, such as care delivery, nursing, administration, and support. Thus, to reduce operating costs by improving energy efficiency and reducing environmental pollutants throughout the work process, detailed plans are necessary. Moreover, organizational performance may be improved by enhancing the public image of the healthcare organization, customer satisfaction, and reducing operating costs, all of which may be induced by the establishment of a green healthcare environment. From a sustainability perspective, customer satisfaction is an important factor, one that is directly related to the quality of the medical services delivered, the operational expenses, and efforts to ensure that customer expectations are met [69,70]. Therefore, continuous green healthcare improvement activities may positively affect the performance of organizations. Based on the above discussions, the following hypothesis is proposed:

**Hypothesis** **7** **(H7).**
*Continuous improvement activities related to green healthcare have a positive effect on environmental performance.*


### 3.2. Operational Definition of Variables

In this paper, green healthcare is defined as an environmentally sustainable approach to hospital operations and methods of delivering medical services. Successful implementation of green healthcare requires contributing factors such as constructing energy-efficient buildings, using eco-friendly products, and decreasing waste and energy use [14,20,33].

The role of top management was defined by the level of its willingness and support of implementing green healthcare. Top management should be engaged in actions that set appropriate visions and goals, build an organizational culture, and motivate employees by providing continuous education and training opportunities while constantly monitoring activities [19,20,33].

The participation of employees is defined as the degree of employees’ positive participation in activities to implement green healthcare. Although there is a great variety in healthcare positions and the individual ability to engage in green healthcare practices, which differs by the nature of each work, the active participation of all employees is a prerequisite to achieving organizational goals [22].

The education and training of employees can be defined as opportunities to gain relevant expertise, learn concepts, and practice methods that will aid the work performance of all those participating in green healthcare implementation [22]. For this, analyses and evaluations of the effectiveness and practicality of the education and training opportunities provided by the healthcare organization should be conducted.

Green healthcare monitoring activities represent the degree to which the management and supervisory personnel collect data regarding the efficacy of green healthcare activities, such as quantities of hazardous chemicals disposed of, recycling, and reducing the waste volume. To ensure the success of green healthcare, it is necessary to establish an appropriate healthcare working environment where monitoring can be conducted seamlessly. Furthermore, flexibility must be incorporated to adequately respond to the changing organizational and environmental conditions [71].

Continuous green healthcare improvement activities are defined as organizational efforts to reflect all changes in the environment and customer demands related to the delivery of care services. The implementation of green healthcare requires the long-term commitment of all employees, denoting the need to continuously improve the related processes [20].

Environmental performance is defined as financial and non-financial outcomes achieved through green healthcare practices. In terms of finances, operating costs can be reduced through waste reduction and improving energy efficiency. The non-financial performance can be improved by enhancing customer (i.e., patient) satisfaction through eco-friendly operations, reducing environmental hazards in local communities through the provision of a safe and efficient medical environment, and performing proper corporate social responsibility (CSR) goals [9,17,69,70].

In this study, the measurement items of green healthcare are based on those developed by previous studies and our own work, as shown in Table 2.

## 4. Research Methodology

### 4.1. Data Collection

This study collected data from tertiary hospitals (with more than twenty medical specialty departments) and general hospitals (generally with more than nine medical specialty departments) in South Korea. We chose those hospitals as “small hospitals often do not share the complexity issues of large hospitals and may not have developed extensive quality management systems” [73].

A survey questionnaire was developed initially in English and then it was translated into Korean by two bilingual operations management faculty using the double translation protocol [74]. The initial questionnaire was tested by nursing managers, medical technicians, and administrators in thirty Korean hospitals as a pilot test to review whether the questionnaire items accurately and fully explained our research questions, and then refined or eliminated some items suggested by the subjects because of ambiguity or difficulty to measure certain items precisely. The Korean version of the questionnaire was translated back into English by two bilingual faculties in the service quality area. The two English versions of the questionnaire had no significant difference.

Data were collected from the staff of the selected Korean hospitals from May 25 to 10 July 2021. Hospitals in this survey participated on a voluntary basis. Out of the 1000 questionnaires that were distributed to employees in these hospitals, we received 276 (27.60%) responses. Fifteen incomplete questionnaires were discarded. The final sample consisted of 261 (26.10%) valid questionnaires. Table 3 summarizes the sample profile. As shown in Table 3, 13.4% of respondents were with public hospitals and 86.6% with private hospitals. The classification types of hospitals represented were tertiary (55.9%) and general (44.1%). The number of beds of hospitals ranged from more than 160 to more than 1000. The study participants’ positions included managers (24.9%), team leaders (26.4%), and front-line employees (48.7%). The proportion of respondents in decision-making positions was 51.3%, which is considered an appropriate sample for this study. The respondents’ occupations included: nurse (33.0%), medical technician (26.4%), administrator and physician (16.1% each), and pharmacist (8.4%).

As shown in Table 4, in response to the question of whether there is a department in charge of green healthcare activity in the hospital, 7.7% affirmative, 10.0% progressing toward the department, 26.0% under discussion, and 42.5% not sure. Although a few of the surveyed hospitals had an active department in charge of green healthcare, the participants had a varying degree of perceptions about the current implementation state of green healthcare activities: well (17.2%), average (57.5%), and no (25.3%).

The surveyed hospitals were implementing green healthcare with the following activities (multiple checks): reduction of hospital waste discharge (55.6%), energy consumption reduction (55.2%), reduction of infectious medical waste discharge (46.0%), reducing food waste discharge (42.1%), reduced use of PVC (21.8%), purchasing eco-friendly products (17.6%), and purchasing low-carbon food (14.6%).

In our research sample, the proportion of employees who participated actively in green healthcare was low at 13.4%, the number of staff interested in the program was 25.3%, and those who recognize the need for the program was 12.6%. These results imply that, although many issues related to green healthcare campaigns are widely publicized, hospital employees generally have a low level of awareness of or willingness to participate in green healthcare activities in their organizations.

### 4.2. Model Variables

The questionnaire used 5-point Likert scales to measure the constructs of the study. The data was analyzed by SPSS 23.0 (IBM, New York, NY, USA) and the AMOS 23.0 (IBM, New York, NY, USA) programs for structural equation modeling (SEM), which provide all of the tools necessary to test the hypotheses. Reliability was tested based on Cronbach’s alpha value (Table 5). All of the coefficients of reliability measures for the constructs exceeded the threshold value of 0.70 for basic research [75]. In the reliability test, Cronbach’s alpha value for continuous improvement activities was the highest (0.956), while education and training was the lowest (0.917). All of the Cronbach’s alpha values for the six latent variables were greater than 0.70. Composite reliability (CR) is considered to be a less biased estimate of reliability than Cronbach’s alpha, and the acceptable value of CR is above 0.70 [76].

To test the validity of the accuracy for a measure, confirmatory factor analysis (CFA) was performed to identify the most meaningful basis and to examine similarities and differences of the data based on Brown’s [77] recommendation. To provide evidence of the convergent and discriminant validity of theoretical constructs, CFA was employed to test measurement models for each construct. As the results of CFA, the values of standardized regression weights of all the variables proposed by the study were shown to be greater than 0.7 and statistically significant at the 0.05 level (see Table 5). AVE, which measures the level of variance captured by a construct versus the level due to measurement error, above 0.70 would be considered a very good acceptable value [76]. The values of AVE and CR for the role of top management, participation of employees, education and training, monitoring activities/systems, continuous improvement activities, and environmental performance were all greater than 0.70 and 0.90, respectively.

The research model consisted of six major components, which were measured by observed variables: the role of top management, participation of employees, education and training, monitoring activities/systems, continuous improvement activities, and environmental performance. The results of the goodness of fit tests for the measurement models are summarized in Table 6. Compared to the recommended values for the goodness of fit tests, the values of CFI, RMR, SRMR, RMSEA, and χ^2^/d.f. of the measurement models were satisfactory, while the value of GFI was not (0.810).

Table 7 provides the square roots of average variance extracted (AVE) of latent variables, while the off-diagonal elements are correlations between latent variables. For discriminant validity, the square root of AVE of any latent variable should be greater than the correlation coefficient between this particular latent variable and other latent variables, and correlation between variables is not high (less than 0.8) [76,78]. The statistics shown in Table 7 satisfied this requirement, lending evidence to discriminant and construct validity.

## 5. Results and Discussion

The goodness of fit test was used to assess hypotheses for the proposed research model. The model’s values of CFI (0.958), RMR (0.028), SRMR (0.025), and RMSEA (0.071) indicated good fit, and χ^2^/d.f (2.274) was significant. However, the value of GFI (0.836) did not meet the acceptable value.

This study needs to control for hospital size to remove any compounding effects in the analysis. Thus, we included a control variable of hospital size in the model to check its effect. Hospital size could potentially be a confounding variable contributing to the role of the top management. Large hospitals tend to have more complex organizational structures, work manuals, and modern facilities, equipment, and systems than smaller hospitals [63]. However, the results of our proposed models showed no significant effect of hospital size on the model outcome.

Table 8 presents the results of hypotheses tests. For H1, H2, and H3, the standardized path coefficient between the role of top management and participation of employees (H1) was 0.887 and the coefficient between the role of top management and education and training (H2) was 0.863. The coefficient between the role of top management and monitoring activities/systems (H3) was 0.941. These coefficients are statistically significant at the 0.001 level, supporting H1, H2, and H3. The results imply that the role of top management has positive influences on the participation of employees, education and training, and monitoring activities/systems for green healthcare. The results confirm that top management can foster employees’ motivation by providing and delivering meaningful leadership for green healthcare [19,34]. As emphasized in previous studies (e.g., [14,33]) or by associations/institutions (e.g., [7,47]), top management of healthcare organizations should provide the leadership for green practices by providing various encouragements, resources, and incentives for employee participation in green activities.

For H4, the standardized path coefficient between the participation of employees and continuous improvement activities was 0.209, not statistically significant at the 0.001 level. Thus, H4 was not supported. Previous studies have suggested that the participation of employees positively impacts continuous improvement activities. However, the results of our study were contrary to that of previous studies. The following condition is most likely to explain the contradicting results. While many employees may be willing to participate in green healthcare activities, they appear to be skeptical about the impact of their individual green activities on the environment. Thus, organizations should show employees with proven evidence (e.g., indicators and/or economic impacts) that every seemingly minute green activity can contribute to environmental protection. For H5 and H6, the standardized path coefficients between continuous improvement activities and education and training (H5) and monitoring activities/systems (H6) were 0.364 and 0.940, respectively, and both statistically significant at the 0.001 level, thus supporting the two hypotheses. These results are consistent with those of previous research [20,22]. Thus, green healthcare education, training, and monitoring of activities/systems can efficiently and effectively develop continuous improvement activities using green concepts, procedures, and/or systems.

For H7, the standardized path coefficient between continuous improvement activities and environmental performance (H7) was 0.963 and statistically significant at the 0.001 level, thus supporting H7. This result implies that continuous improvement activities influence long-term environmental performance [20,64]. Green healthcare activities of hospitals can not only contribute to environmental sustainability by minimizing pollutants across their healthcare services, but they can also enhance their public image as socially responsible organizations. In addition, hospitals can reduce their operating costs through continuous green healthcare improvement activities [20,33,64]. As a result, healthcare institutions should strive for long-term green healthcare efforts in order to achieve better organizational performance.

The results of our study indicate that for a hospital to effectively implement TQM practices for green healthcare implementation, the role of top management is critical to encourage employees’ active participation in the program, education and training, monitoring of the activities, continuous improvement, and environmental performance. Thus, the development of green hospitals requires an active commitment to and participation in green healthcare activities by top management [33]. In addition, the study results imply that sampled hospitals are efficiently implementing TQM practices, which can enhance environmental performance. Therefore, to initiate green systems or processes in the TQM perspective, a hospital needs to develop requisite structures and culture to achieve environmental performance. Examples of green healthcare activities in the realm of TQM include: green hospital policy, implementation and maintenance of clear work guidelines, development of environmental programs, purchase of eco-friendly products, reduction of energy consumption, lowering waste emissions, and reducing the consumption of harmful and toxic substances [14,33,61].

Many hospitals use systems that combine software and manual processes to develop green healthcare. Some systems or processes focus on green healthcare without the participation of organization members [32]. However, for effective green healthcare management, hospital managers should engage the front-line employees who can actually reduce waste, provide timely information on green programs, identify cost-saving opportunities, and provide clear outcome of informed decisions on green healthcare for celebration. Thus, hospitals should implement green activities that contribute to the greater good through unique and creative processes through active employee engagement.

## 6. Conclusions

This study tried to provide insights on the importance of employee participation for implementing effective green healthcare programs based on the perspective of TQM. In addition, this study investigated the role of top management, participation of employees, education and training of employees, the monitoring activities of employees, continuous improvement activities, and environmental performance for the realization of green healthcare.

The results of the hypotheses tests confirmed the effects of top management’s role on the participation of employees (H1), education and training (H2), and monitoring activities/systems (H3). The results of this study show that the leadership of top management in healthcare organizations is imperative for the successful implementation of green healthcare. Top management’s inspiring commitment to green healthcare can motivate employees’ active participation in supportive activities and related education and training programs, thus facilitating successful implementation of green healthcare operations.

The study results also revealed positive relationships between continuous improvement activities and education and training (H5) and monitoring activities/systems (H6). The study also confirmed the positive effect of continuous improvement activities on environmental performance (H7). The results of this study demonstrate that green healthcare activities can only be successful with active participation of all organization members in continuous education/training, monitoring, and feedback for the program.

However, this study found no positive relationship between participation of employees and continuous improvement activities (H4). This result indicates that it is necessary to develop strategies to encourage employee participation and engagement in green healthcare activities. That implies that simple encouragement of employees to participate in green activities on a voluntary basis would have very limited effects. Therefore, the hospital management needs to establish operational strategies that can positively motivate its employees to actively participate and engage in green healthcare activities.

To discern the reasons for the nonsupport of Hypothesis 4, we discussed the issue with four managers of the medical staff (i.e., doctor, nurse, and medical technician) and supporting department (i.e., administrator) at sample hospitals. The most likely reasons for this result based on their judgment are as follows: First, Korean hospitals must strictly adhere to government regulations regarding the handling of medical waste. The Waste Management Act (Law No. 17851) prohibits hospitals from including infectious medical waste in their general waste for disposal. To make sure that the regulation is not violated, hospital personnel tend to include all waste, including the general waste, as infectious medical waste for disposal. While this is a chronic problem of waste management in hospitals, it reflects the general perception of hospital employees that their active participation in green healthcare would have no real impact on the overall effectiveness of the program. Second, green healthcare initiatives were launched as event-based activities with the support of the Korean government in the early 2010s with some early achievements. However, when the government discontinued its financial support, hospitals were burdened with funding their own green healthcare programs, which had no short-term visible results. Subsequently, many hospitals did not find compelling motivation to continue green healthcare activities, primarily due to financial reasons as well as hospital employees’ perception that their participation in the program had no visible outcomes. Consequently, employees did not feel motivated to continuously improve green healthcare activities, as there was no strong leadership support from either the government or the hospital management. Third, there were few incentives to implement green healthcare activities, as no specific department or unit was charged with exclusive responsibilities for the program. Fourth, no collaborative arrangement with external partners was observed (e.g., university research centers, professional societies, task forces of government departments, etc.) for learning, training, and recognition of opportunities. Lastly, the recent strategic emphases of hospitals tend to focus on urgent current needs such as treating COVID-19 patients and the implementation of ICT-related programs. Thus, in the absence of strong leadership or support of the government, hospitals consider green healthcare as a nice to have program rather than an imperative one. In addition, to sustain the program, employees need to be constantly encouraged and informed that their small contributions can have significant combined effects on green healthcare practices.

### 6.1. Theoretical and Practical Implications

Although there is no research on quality management-based green healthcare for medical institutions, this study contributes theoretical and practical in terms of suggesting the possibility of approaching from the perspective of TQM as a method for implementing green healthcare. First, the results of our study offer important theoretical implications. As this study represents the first effort to combine the TQM perspective with operational plans of green healthcare, its academic value lies in expanding the research area of sustainability in healthcare. Second, this study identified key factors that are essential for the successful implementation of green healthcare in an organization, such as active leadership of top management, motivating the participation of employees in green activities, education and training, and continuous improvement of the program. Third, the findings of this study could be used as a basis for future research on effective management of green healthcare and its effect on organizational performance.

The results of this study have several practical implications. First, the findings of this study could be used as a good starting point for developing best practices of green healthcare activities. Second, the results could be used to develop strategic plans for successfully aligning different roles of the medical staff, administration departments, and external stakeholders in developing and implementing green healthcare activities. For example, externally, the cooperation of all stakeholders, such as eco-friendly construction companies and suppliers, visitors, and local/state government officials, could be enhanced and strengthened for the greater good. Third, the findings of this study provide understanding and knowledge about the required antecedents for achieving green healthcare (e.g., participation activities, education and training, and monitoring). Finally, the environmental performance metrics developed in this study can be used for campaigns that are aimed at supporting the practice of green healthcare among healthcare organizations. Such campaigns can foster cooperation among all stakeholders and communities to promote green healthcare programs [79].

### 6.2. Limitations and Future Research Directions

This study has several limitations. First, the study data were collected from tertiary and general hospitals in South Korea with more than 160 beds. Although the green healthcare movement in South Korea has been around for a long time, its implementation among the hospitals has been quite varied. Second, this study did not investigate unique aspects of green healthcare activities in each participating hospital, assuming that the program would be similar among the sampled hospitals. The characteristics of the type of green healthcare activities of each hospital might have some influence on the study results. Lastly, the measurement items of this study were based on the perception of employees. Thus, the generalizability of the results of this study would be limited. The limitations of this study described above can provide opportunities and directions for the future research in the green healthcare area. Additionally, a cross-cultural and longitudinal study would provide robust insights on green healthcare programs.

## Figures and Tables

**Figure 1 ijerph-19-06504-f001:**
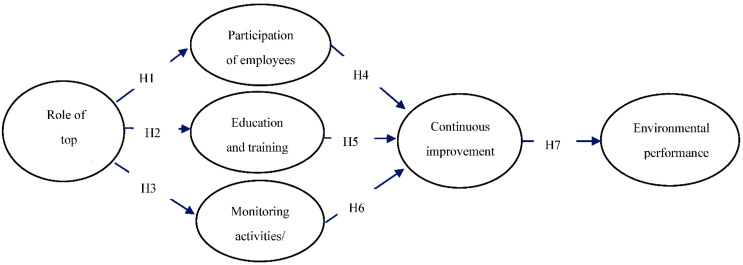
Proposed research model.

**Table 2 ijerph-19-06504-t002:** Measurement items.

Component	Measurement Items	References
Role of top management	RTM1: Develop and commit to a system-wide green hospital policy RTM2: Form a task force consisting of representatives of various departments and professions within the hospital to help guide and implement strategies RTM3: Ensure that strategic and operating plans and budgets reflect the commitment to a green hospital RTM4: Set green goals for employees RTM5: Create a work environment which is conducive for employee engagement in green activities	[33,61]
Participation of employees	PEA1: Degree of employee participation in the practice of green healthcare PEA2: Reflection of employees’ opinions in decision making PEA3: Degree of employee cooperation to achieve goals PEA4: The green work I do is meaningful to me	[61,72]
Education and training	EAT1: Provide employees with training to develop their knowledge and skills required for green healthcare EAT2: Provide employees with training to promote green values EAT3: Evaluation of the effectiveness of education and training EAT4: Provision of human and material resources required for education and training	[33,61]
Monitoring activities/systems	MAS1: Development and maintenance of clear work guidelines for monitoring green healthcare activities MAS2: Assurance of an appropriate level of work related to the monitoring green healthcare activities MAS3: Ensuring compliance with environmental regulations and the required degree of monitoring for operations MAS4: Continuous monitoring of green healthcare activities	[14]
Continuous improvement activities	CIA1: Continuous reassessment and revisions (when necessary) of green healthcare activities CIA2: Improvement of operational plans to enhance green healthcare activities CIA3: Development of programs to improve green healthcare activities CIA4: Reflection of customer requirements for continuous improvement	[14]
Environmental performance	EPE1: Lower use of water in our facilities than that during the pre-green healthcare practices period EPE2: Reduction in energy (power) use in our facilities compared with that during the pre-green healthcare practices period EPE3: Lower consumption of harmful and toxic substances in our facilities than that during the pre-green healthcare practices period EPE4: Lower waste emissions in our facilities than those during the pre-green healthcare practices period EPE5: Purchase of eco-friendlier products than that during the pre-green healthcare practices period in our facilities	[61,72]

**Table 3 ijerph-19-06504-t003:** Hospital characteristics and respondents’ demographic data.

Employees Respondents’ Characteristics			Hospitals’ Characteristics		
Items		Frequency (Percent)	Items		Frequency (Percent)
Gender	Male	74 (28.4%)	Hospital type	Tertiary hospitals	146 (55.9%)
	Female	187 (71.6%)		General hospitals	115 (44.1%)
Age	20s 30s 40s 50s 60s	34 (13.0%) 71 (27.2%) 77 (29.5%) 75 (28.8%) 4 (1.5%)	Ownership	Private hospital	226 (86.6%)
				Public hospital	35 (13.4%)
			Number of beds	160 to 300	34 (13.0%)
				301 to 500	18 (6.9%)
				501 to 1000	144 (55.2%)
				1001 more	65 (24.9%)
Position	Manager Team Leader Front-line employee	65 (24.9%) 69 (26.4%) 127 (48.7%)	Location	Metropolitans	200 (76.6%)
				Provinces	61 (23.4%)
Position	Nurse Medical technician Administrator Physician Pharmacist	86 (33.0%) 69 (26.4%) 42 (16.1%) 42 (16.1%) 22 (8.4%)	In South Korea, ● Private hospitals operated by universities, corporations, medical corporations, or individual. ● Public hospitals operated by government support. ● Hospital classification type: a tertiary general hospital, a secondary general hospital, and a hospital.		
Total number of respondents		261 (100.0%)			

**Table 4 ijerph-19-06504-t004:** Green healthcare activities of korean hospitals.

Items	Sub-Items	Frequency (Percent)
Our hospital has a department in charge of green healthcare activity	Yes, our hospital has the department	20 (7.7%)
	Just progressing the department	26 (10.0%)
	Just being discussed about the department	68 (26.0%)
	Not interested in having that department	36 (13.8%)
	Not sure	111 (42.5%)
Our hospital is implementing green healthcare activities	It is very much so	45 (17.2%)
	It seems to run on average.	150 (57.5%)
	No	66 (25.3%)
Green healthcare activities of our hospital include (multiple checks):	Reduction of hospital waste discharge	145/261 (55.6%)
	Energy consumption reduction	144/261 (55.2%)
	Reduction of infectious medical waste discharge	120/261 (46.0%)
	Reduce food waste discharge	110/261 (42.1%)
	Reduce PVC use	57/261 (21.8%)
	Purchasing eco-friendly products	46/261 (17.6%)
	Purchasing low-carbon food	38/261 (14.6%)
The overall atmosphere of our hospital’s green healthcare activities	An atmosphere in which all employee actively participates	35 (13.4%)
	Only interested employee participates	66 (25.3%)
	Employee’s interest is low	49 (18.8%)
	An atmosphere that only recognizes the need	33 (12.6%)
	Not sure	78 (29.9%)
Total number of respondents		261 (100.0%)

**Table 5 ijerph-19-06504-t005:** Results of Cronbach’s alpha, AVE, composite reliability, and CFA.

Constructs	Variables	Standardized Loading	*t*-Value	*p*-Value	Cronbach’s Alphas	AVE	CR
Role of the top management	RTM1	0.831	18.775	0.000 0.000 -	0.951	0.778	0.946
	RTM2	0.848	19.585				
	RTM3	0.867	20.577				
	RTM4	0.879	21.224				
	RTM5	0.899	-				
Participation of employees	PEA1	0.836	18.929	0.000	0.920	0.794	0.939
	PEA2	0.835	18.914	0.000			
	PEA3	0.881	21.259	0.000			
	PEA4	0.890	-	-			
Education and training	EAT1	0.878	16.395	0.000	0.917	0.767	0.929
	EAT2	0.872	16.255	0.000			
	EAT3	0.903	17.067	0.000			
	EAT4	0.787	-	-			
Monitoring activities/systems	MAS1	0.892	23.022	0.000	0.939	0.826	0.950
	MAS2	0.888	22.739	0.000			
	MAS3	0.875	21.970	0.000			
	MAS4	0.910	-	-			
Continuous improvement activities	CIA1	0.894	14.583	0.000	0.956	0.877	0.966
	CIA2	0.929	13.706	0.000			
	CIA3	0.919	14.586	0.000			
	CIA4	0.934	-	-			
Environmental performance	EPE1	0.919	22.642	0.000	0.947	0.786	0.948
	EPE2	0.933	23.520	0.000			
	EPE3	0.830	19.276	0.000			
	EPE4	0.845	18.184	0.000			
	EPE5	0.882	-	-			

CR (Composite Reliability) = ∑ (factor loading) 2/[∑ (factor loading)2 + ∑ (error)] AVE=∑ (factor loading) 2/[∑ (factor loading)2 + ∑ (error)].

**Table 6 ijerph-19-06504-t006:** Results of fit indices for CFA.

	χ^2^	d.f	χ^2^/d.f	GFI	CFI	RMR	SRMR	RMSEA
Measurement model	738.017	279	2.645	0.810	0.947	0.031	0.027	0.080
Recommended values			≤3.0	≥0.9	≥0.9	≤0.08	≤0.08	≤0.08

GFI: goodness of fit index CFI: comparative fit index RMR: root mean square residual SRMR: standardized root mean square residual RMSEA: root mean square error of approximation.

**Table 7 ijerph-19-06504-t007:** Correlation matrix and average variance extracted (AVE).

Factor	Role of Top Management	Participation of Employees	Education and Training	Monitoring Activities/Systems	Continuous Improvement Activities	Environmental Performance
Role of top management	**0.881**					
Participation of employees	0.769	**0.893**				
Education and training	0.743	0.702	**0.875**			
Monitoring activities/systems	0.752	0.754	0.711	**0.908**		
Continuous improvement activities	0.661	0.630	0.795	0.730	**0.936**	
Environmental performance	0.629	0.614	0.773	0.796	0.703	**0.886**
AVE	0.778	0.794	0.767	0.826	0.877	0.786

Bold value is the square root of AVE.

**Table 8 ijerph-19-06504-t008:** Results of Hypotheses Tests.

Path			Path Coefficient	S.E.	*t*-Value	*p*-Value	Hypothesis Test
Role of top management	⟶	Participation of employees	0.887	0.045	21.892	0.000 *	Supported H1
Role of top management	⟶	Education and training	0.863	0.047	20.038	0.000 *	Supported H2
Role of top management	⟶	Monitoring activities/systems	0.941	0.057	15.052	0.000 *	Supported H3
Participation of employees	⟶	Continuous improvement activities	0.209	0.155	1.505	0.132	Not Supported H4
Education and training	⟶	Continuous improvement activities	0.364	0.086	3.842	0.000 *	Supported H5
Monitoring activities/systems	⟶	Continuous improvement activities	0.940	0.133	7.585	0.000 *	Supported H6
Continuous improvement activities	⟶	Environmental performance	0.963	0.041	21.759	0.000 *	Supported H7

* *p* < 0.001.

## Data Availability

Not applicable.

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
