# Peer review of "Developing Green Healthcare Activities in the Total Quality Management Framework"

_ijerph, 2022, doi:10.3390/ijerph19116504_

Round 1
Reviewer 1 Report
Dear Authors,
Thanks for submitting this manuscript, which addresses a very relevant and timely topic. I enclose some suggestions for improving your paper, which, in my opinion, reads very well.
In the Introduction, I would clearly state one (or more) research question(s).
In the Theoretical Framework, I would include one specific paragraph dedicated to TQM, highlighting the main academic trends recently debated on this issue. I would include this paragraph as section 2.2., moving the paragraph “Green Healthcare and TQM” to section 2.3.
I would better organize the concluding sections of your work. First, I would include a “Discussion” section, where to interpret your findings also on the basis of what was already known on the topic (the theoretical framework). Then, I would move to conclusions, where to clearly answer to the research question(s) raised in the introduction, as well as to include considerations related to limitation, practical/policy implications of your analysis and directions for future research.
Best wishes!
Author Response
Response to the Comments:
Thank you for these comments. We do agree with the reviewer’s view about the TQM concept.
Please see attached file.

Reviewer 2 Report
I have carefully read the paper, that is well structured and written. The introduction explains well the aim of the paper and the literature review section depicts in a rich way the framework for the research. The methodology is robust and well defined. Hypotheses are well explained. I only suggest to Authors to better discuss the results in comparison with previous literature analyzed, since after the presentation of results they skip to conclusions and implications without discussing the results comparing them with previous literature.
Author Response
Response to the Comments:
Thank you very much for your positive comments on our paper. Based on the reviewer’s suggestions, we added the following in the Conclusion section.
p.15:
Many hospitals use a system that combines software and manual processes to develop green healthcare. Some systems or processes focus on green healthcare without the participation of organization members. However, for effective green healthcare management, hospital managers should engage the front-line employees who can eliminate waste, provide timely information, identify cost-saving opportunities, and provide visibility to make informed decisions throughout their daily activities. Thus, the development of green hospitals requires an active commitment to and participation in green healthcare activities by top management [33]. Although there is no research on quality management-based green healthcare for medical institutions, this study tried to provide insights on the importance of employee participation for implementing effective green healthcare programs based on the perspective of TQM. In addition, this study investigated the role of top management, participation of employees, education and training of employees, the monitoring activities of employees, continuous improvement activities, and environmental performance for the realization of green healthcare.
Reviewer 3 Report
Interesting publication on a very current and important problem - Total Quality Management Framework. The work was prepared very carefully, taking into account the scientific nature. The work contains a review of the literature, which is a good introduction to the research part. The research goals have been correctly defined and correspond to the research hypotheses. In the analysis, the authors use the results of the conducted surveys. The conclusions were developed on the basis of in-depth statistical analyzes. A great advantage of the publication is the part containing statistical material as well as implications and proposed directions of future research. The published publication meets the requirements of a scientific article.
Author Response
Response to the Comments:
Thank you very much for your favorable comments on our paper.
Round 2
Reviewer 1 Report
Dear Authors,
Thanks for submitting the revised version of your work, which, however, does not address all the comments of the first review.
I would again suggest you to include a "Discussion" section, just after "Results" and before "Conclusions".
Best wishes!
Author Response
Response to Reviewers’ Comments
We greatly appreciate the valuable suggestions by the reviewer. We highlighted our changed parts in the revised paper, based on the reviewers’ comments using yellow color.
Please find the attached file.
